# Application of elementary probability models for text homogeneity and segmentation: A case study of Bible

Berhane Abebe [1,2]*

**1** Department of Statistics, MaiNefhi College of Science, Mai-Nefhi, Zoba Maekel, Eritrea, **2** Department of Probability Theory and Mathematical Statistics, Mathematics and Mechanics Faculty, Novosibirsk State University, Novosibirsk, Russia Federation

* b.andemikael@g.nsu.ru

**Data Availability Statement:** You can access the minimum data from the public repository https://figshare.com/s/10cde46cff91bd64e1e2.

**Funding:** The author received no specific funding for this work.

## Abstract

For the purpose of this study, A statistical test of Biblical books was conducted using the recently discovered probability models for text homogeneity and text change point detection. Accordingly, translations of Biblical books of Tigrigna and Amharic (major languages spoken in Eritrea and Ethiopia) and English were studied. A Zipf-Mandelbrot distribution with a parameter range of 0.55 to 0.88 was obtained in these three Bibles. According to the statistical analysis of the texts' homogeneity, the translation of Bible in each of these three languages was a heterogeneous concatenation of different books or genres. Furthermore, an in-depth examination of the text segmentation of prat of a single genre—the English Bible letters revealed that the Pauline letters are heterogeneous concatenations of two homogeneous segments.

## 1 Introduction

A number of studies have been conducted so far to demonstrate the frequency of words and the rank order of a text document follows a Zipfian or near-zipfian distribution. Zipf's distribution can be seen for language families with a common ancestor as well as for several languages with various exponents [1]. A random simulation of randomly generated text following Zipf's distribution was first suggested by [2], but then rejected by [3]. Similar studies have been conducted in three languages, namely: English, French, and German. Based on the corpus study, results revealed that Zipf's law and its derivatives fits the data well in a certain range text length [4].

Numerous probabilistic models have been discovered to create statistical tests of natural language based on Zipf's law and its derivations. For instance, an elementary probabilistic model was used by [5] to determine whether the frequency of terms in text follows a Zipfian or near-Zipfian distribution. Prior to that, [6] investigated the functional central limit theorems of urn models and statistical relationship between Bahadur, Zipf's and Heap's laws. However, in this study elementary probabilistic model based on the notion that words have been arbitrarily chosen from an infinite dictionary was employed. The infinite urn models and this system have comparable structures.

**Competing interests:** The author have declared that no competing interests exist.

Similar to this, a few studies on the statistical analysis of religious texts have been carried out. [4] found that the English and German versions of the Bible almost closely resemble the distribution of general texts, whereas the Hungarian version for a small corpus had a distorted shape. This is alarming, in the sense that religious texts are written in human language and, therefore, expected to have the same distribution as other non-religious texts. A study to interpret the term inheritance in the four Gospels was initiated by [7]. The objective of the study was to explore the primary source of the Gospels. A mathematical framework discovered by [8] demonstrates a high level of text homogeneity. Hence, words with rank frequencies of fewer than 3000–4000 adhere to Zipf-Mandelbrot distribution. In a similar vein, the Gospels have 50% of similarity indexes [8]. [7] split the Gospels into two categories based on the literary styles: John utilizes one style, whereas Matthew, Mark, and Luke are synoptic Gospels. [7] came to the statistically singular conclusion that there is only one single solution: Mark was used as a reference by Matthew and Luke, but Mark used neither Matthew nor Luke as a reference. The author accomplished this by using the double and triple link fork models. Using a novel approach to taxicab correspondence analysis, [5] explored the two theories of the sources of the gospel, such as the two source hypothesis and the two gospel hypothesis. A recent statistical investigation of 100 translations of the Bible into 100 different languages found that the power law exponent value is almost close to one, which is exactly in accordance with Zipf's law [9]. In this study, Bible translations into Tigrigna, Amharic, and English were statistically analyzed for the following reasons: First, to demonstrate that the frequency of words verse rank order follows Zipf-Mandelbrot's distribution, as per [10]. Second, to investigate text homogeneity using the model created by [11] for the difference between forward and backward processes. Finally, and importantly to identify text change points using an elementary probabilistic model of [12].

The paper is organized as follows: Section 2 presents the data sources and the methodology for text homogeneity testing and change point detection. Section 3 includes graphical analysis of text homogeneity and text change point detection. Additionally, other statistical results of parameter estimates were computed. Section 4 includes the homogeneity test, the genre classification of the Bible books and their text change point detection, as well as a detailed examination of one heterogeneous genre in the English Bible. A summary of the study's conclusions is provided in Section 5, while Section 6 includes an acknowledgement.

## 2 Methodology

This section offers highlights of the designations used for variables and parameters as well as the main theoretical background for the derivation of the elementary probability model used for analysis. The latter section, presents details of the methodological section.

- $\theta_n$ is the Zipf's parameter.

- $q_n$ is the Mandelbrot's parameter for modifying the Zipf's law.

- $n$ is the number of words in text document. it is assumed that $n \to \infty$ in theoretical results.

- $R_n$ is total number of different words in a text document of length n.

- $J_n$ is a random variable denoting the distribution of the scaled difference between forward and backward counting processes.

- $T$ is the hypothetical change point moment, where $0 < T < 1$.

- $T_n$ is estimated value of $T$, $0 < T_n < 1$;

- $nT$ is the point of concatenation of texts. We assume that first $\lfloor nT \rfloor$ words and last $n - \lfloor nT \rfloor$ words are from different distributions.

- $nT_n$ is estimated point of text concatenations.

- $\tau$ is the theoretical limit for $\tau_n$ as $n \to \infty$, $0 < \tau < 1$;

    For other notations and designations please refer to [12]

    Zipf's law states that the frequency of a word is a power function of its rank order [13].

$$p_i = c(i)^{-1/\theta}, \quad i \geq 1, \quad 0 < \theta < 1. \tag{1}$$

The [14] modification of the Zipf's Law is given by:

$$p_i = c(i + q)^{-1/\theta}, \quad i \geq 1, \quad 0 < \theta < 1, \quad q > -1. \tag{2}$$

Where $c = (\zeta(1/\theta, q + 1))^{-1}$ and $\zeta(\alpha, x) = \sum_{i=0}^{\infty} (i + x)^{-\alpha}$ is the Hurwitz zeta function. The next Regularity Condition plays an important role in the subsequent derivations:

$$\kappa(x) := \max\{k > 0 : \ p_k \geq 1/x\} = x^\theta L(x), \quad 0 < \theta < 1, \tag{3}$$

$L(\cdot)$ is the slowly varying function of the real argument: $L(tx)/L(x) \to 1$ as $x \to +\infty$ for any real $t > 0$.

$R_n \to$ number of unique words, with $R_0 = 0$.

Let the number of words at the $i$th position be $X_i$ for $1 \leq i \leq n$, with the following assumptions:

$$\mathbf{P}(X_i = j) = p_j > 0, \quad j \geq 1, \qquad p_1 + p_2 + \ldots = 1.$$

[15] proved:

$$\mathbf{E}R_n = \sum_{i=1}^{\infty} (1 - (1 - p_i)^n), \tag{4}$$

$$\mathbf{E}R_n/n \to 0$$

[16] proved the SLLN:

$$R_n/\mathbf{E}R_n \overset{a.s.}{\to} 1.$$

and, if the regular conditions of Eq 3 is fulfilled, then

$$\frac{(R_n - \mathbf{E}R_n)}{\sqrt{\mathbf{E}R_n}} \tag{5}$$

converges weakly to the centered normal distribution with variance

$$2^\theta - 1$$

[6] verified that

$$Z_n = \{Z_n(t), \ 0 \leq t \leq 1\} = \{(R_{[nt]} - \mathbf{E}R_{[nt]})/\sqrt{\mathbf{E}R_n}, \ 0 \leq t \leq 1\} \tag{6}$$

converges weakly in $D(0, 1)$ with uniform metrics to a centered Gaussian process $Z_\theta$ with

continuous a.s. sample paths and covariance function

$$K(s, t) = (s + t)^{\theta} - \max(s^{\theta}, t^{\theta}).$$

The statistical methodology used to analysis and estimate the parameter of the Bible was based on the following three subsection:

## Statistical tests of Zipf-Mandelbrot distribution

The following equations provide parameter estimates for the procedure of counting distinct words in a text document using a Zipf-Mandelbrot's distributions of Eq 2.

$$\widehat{\theta} = \log_2 \frac{R_n}{R_{\lfloor n/2 \rfloor}} \tag{7}$$

$$\widehat{q} = min(q > -1 : r_n(\widehat{\theta}, q) = R_n) \tag{8}$$

$$(r_k) = \sum_{i=1}^{M}(1 - (1 - \widehat{p}_i)^k) + (k\widehat{c})^{\widehat{\theta}} \int_0^{\widehat{kcN^{-\widehat{\alpha}}}} z^{-\theta}e^{-z}dz - N(1 - exp(\widehat{kcN^{-\widehat{\alpha}}}))$$
$$\alpha = \frac{1}{\widehat{\theta}} \quad N = M + 0.5 + \widehat{q}. \tag{9}$$

For detailed derivation see [6]

## Text homogeneity test

The Bible written in three different languages (Tigrigna, Amharic and English) analysed whether each book is homogeneous. Statistical experiment was conducted based on the newly developed Elementary Probability Urn Models. If the regularity condition of 3 holds, the formulas for distribution of scaled difference between forward and backward counting processes, estimation of the parameter, p_value, and the omega square calculations are summarized below:

$$J_n = \frac{\sum_{k=1}^{n}(R_k - R'_k)}{n\sqrt{R_n}} \tag{10}$$

converges weakly to a centered normal random variable with variance $\frac{\theta}{\theta+2}$. Where the p-value is calculated using the tail of the standard normal distribution and the observed value $J_{obs}$ of $J_n$:

$$\mathrm{p-value} = 2\bar{\Phi}\left(|J_{obs}|\sqrt{1 + 2/\theta^*}\right). \tag{11}$$

Where, $\widehat{\theta}$ the estimate of $\theta$ is given by:

$$\widehat{\theta}_n = (\theta_n + \theta'_n)/2. \tag{12}$$

where $\theta_n$ and $\theta'_n$ are the zipf-Mandelbrot's parameters for forward and backward processes

calculated as 7. Then

$$\widehat{\theta}_n = \log_2\left(R_n \Big/ \sqrt{R_{\lfloor n/2\rfloor} R'_{\lfloor n/2\rfloor}}\right), \quad n \geq 2. \tag{13}$$

$$\widehat{W}_n^2 = \frac{1}{3n}\sum_{i=1}^{n-1}\left(\widehat{Z}_n(\frac{k}{n})(2\widehat{Z}_n(\frac{k}{n}) + Z_n(\widehat{\frac{k+1}{n}}))\right) \tag{14}$$

Where $\widehat{W}_n^2$ is the limiting distribution of Eq 10. All these estimates are consistent due to SLLN. For further information [11].

## Text change point detection

The estimated moment of change point detection $T_n$ of $T$ is calculated using [12]. If the intermediate function is given by

$$h_n(t) = t^{\theta_n^*}, \tag{15}$$

so $h_n(0) = 0$, $h_n(1) = 1$. The idea of the intermediate function is to give a function with a strictly increasing limiting function as $n \to \infty$. where the parameter $\theta_n^*$ can be estimated using:

$$\theta_n^* = \frac{\ln\dfrac{\widehat{R}_n(\gamma_n) + \widehat{R}'_n(\gamma_n)}{2}}{\ln\gamma_n}. \tag{16}$$

Hence, the primary and final estimate of the parameter moment of change point T was calculated using Eqs 17 and 18 respectively.

$$\tau_n = \arg\max_{t\in[0,1]}\left(h_n(t) - \min(\widehat{R}_n(t), \widehat{R}'_n(t))\right), \tag{17}$$

$$T_n = (1 - \tau_n)\mathbf{1}\{\widehat{R}_n(\tau_n) > \widehat{R}'_n(\tau_n)\} + \tau_n\mathbf{1}\{\widehat{R}_n(\tau_n) < \widehat{R}'_n(\tau_n)\}. \tag{18}$$

where, $T_n \to T$ in probability.

Percentage error of estimating the text change point detection is computed by:

$$\text{error} = |T_n - T| \times 100\%. \tag{19}$$

For more details on the derivations and proofs of the above formulas see [12].

## Data sources

The Bible books for Tigrigna, Amharic and English has been freely download respectively from the following three websites:

- http://bible.geezexperience.com/tigrigna/download/TigrignaBible.pdf

- http://gzamargna.net/sened_amargnaMetshafQdus.pdf

- https://ebible.org/pdf/eng-asv/eng-asv_all.pdf

## 3 Empirical analysis

### Graphical analysis

The zipf diagram for The Bible translations into three languages Tigrigna, Amharic, and English were presented in Fig 1. The graph demonstrated a power decreasing Zipf-Mandelbrot distribution.

 The graph of the process of counting the number of distinct words in a text (blue line) and its estimate using Zipf-Mandelbrot's (red line) was given in Fig 2. From the graph of the forward process, it is evident that there is heterogeneity, which shows differently in various languages. There are numerous breaks in the power line visible, including two significant ones. The first break, which is more noticeable for the Tigrigna and Amharic languages, located with the start of the Psalms. The second break, which starts at the beginning of the Gospel of John is recognized in all languages, but it is most evident in English. Many new words that were not in the prior text now emerge in these locations, which are connected to the breaks in the power line. In addition, the forward process and its estimate lines almost coincided, indicating that Zipf-Mandelbrot is a superior model for estimating the number of different words in a text.

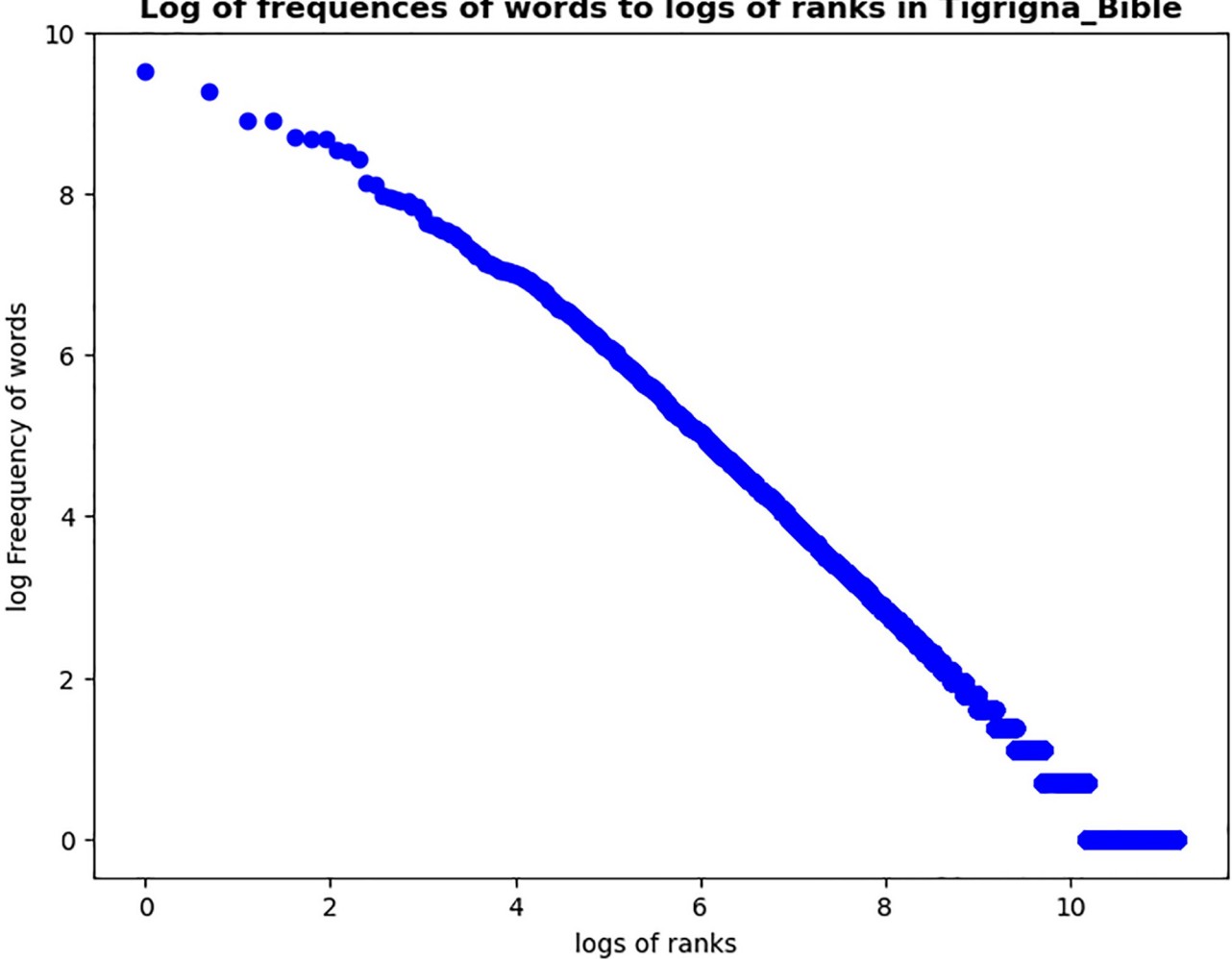

**Fig 1. Zipfian diagrams.** Log-Log graph of Frequencies Vs Rank Order.

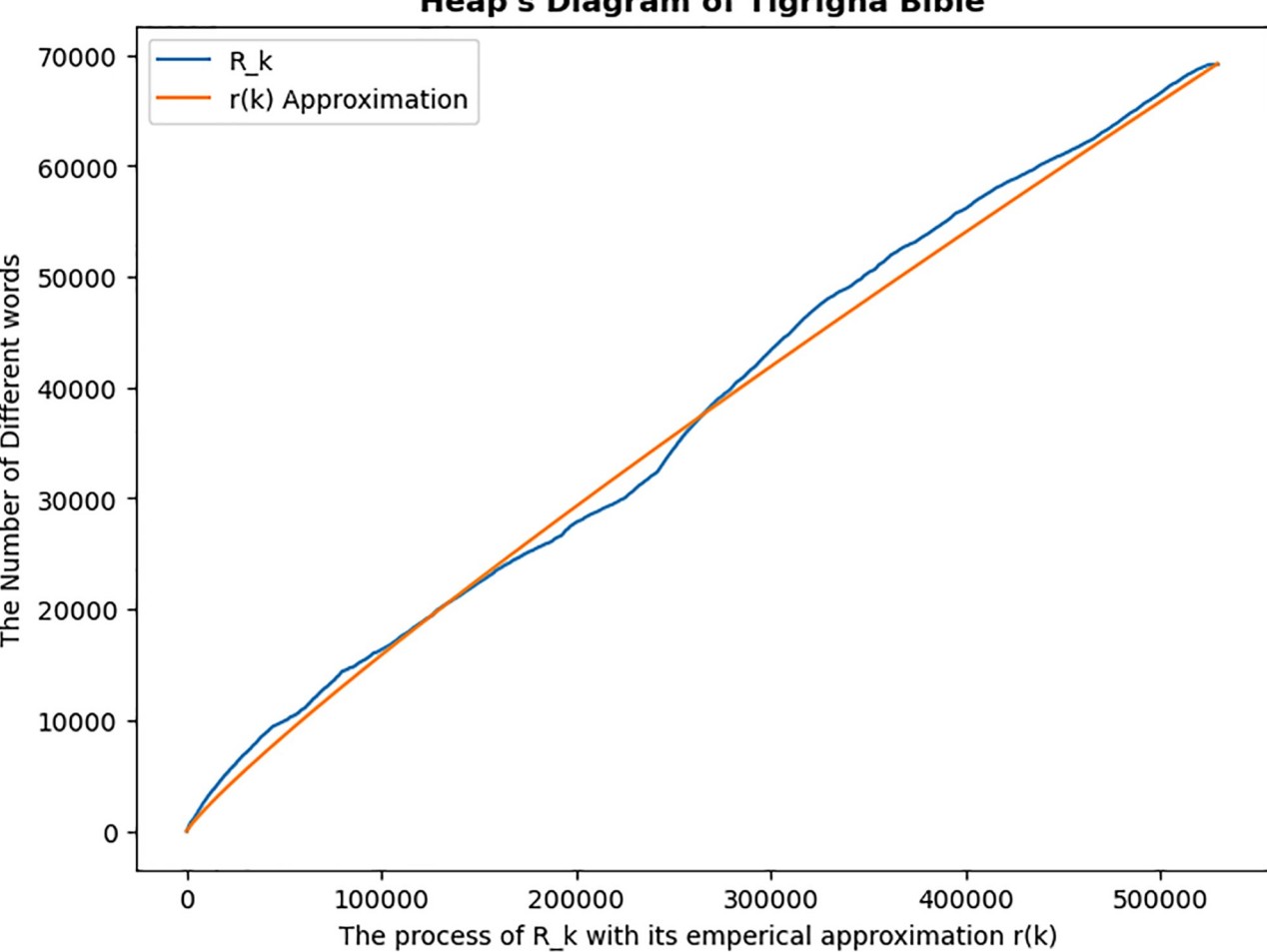

**Fig 2. Heap's diagram.** The blue line represents the graph of word counts versus distinct word counts in a text, whereas the red line represents the graph of its estimated distinct word counts using the Zipf-Mandelbrot method.

The forward process is the process of counting the number of distinct words when reading a document from the beginning to the end. Counting from the end of a written document, on the other hand, is known as the backward process.

The following example was used to demonstrate the forward and backward processes of different words. Punctuation marks have been removed from the study and all words have been changed to lowercase.

Better to have loved and lost, than never to have loved at all:

The format reading for the forward process of counting number of different words after necessary modification is:

better to have loved and lost than never to have loved at all

$k$ 0 1 2 3 4 5 6 7 8 9 10 11 12 13

$R_k$ 0 1 2 3 4 5 6 7 8 8 08 08 09 10

Whereas, the backward process is:

all at loved have to never than lost and loved have to better

$k$ 0 1 2 3 4 5 6 7 8 9 10 11 12 13

$R_k$ 0 1 2 3 4 5 6 7 8 9 09 09 09 10

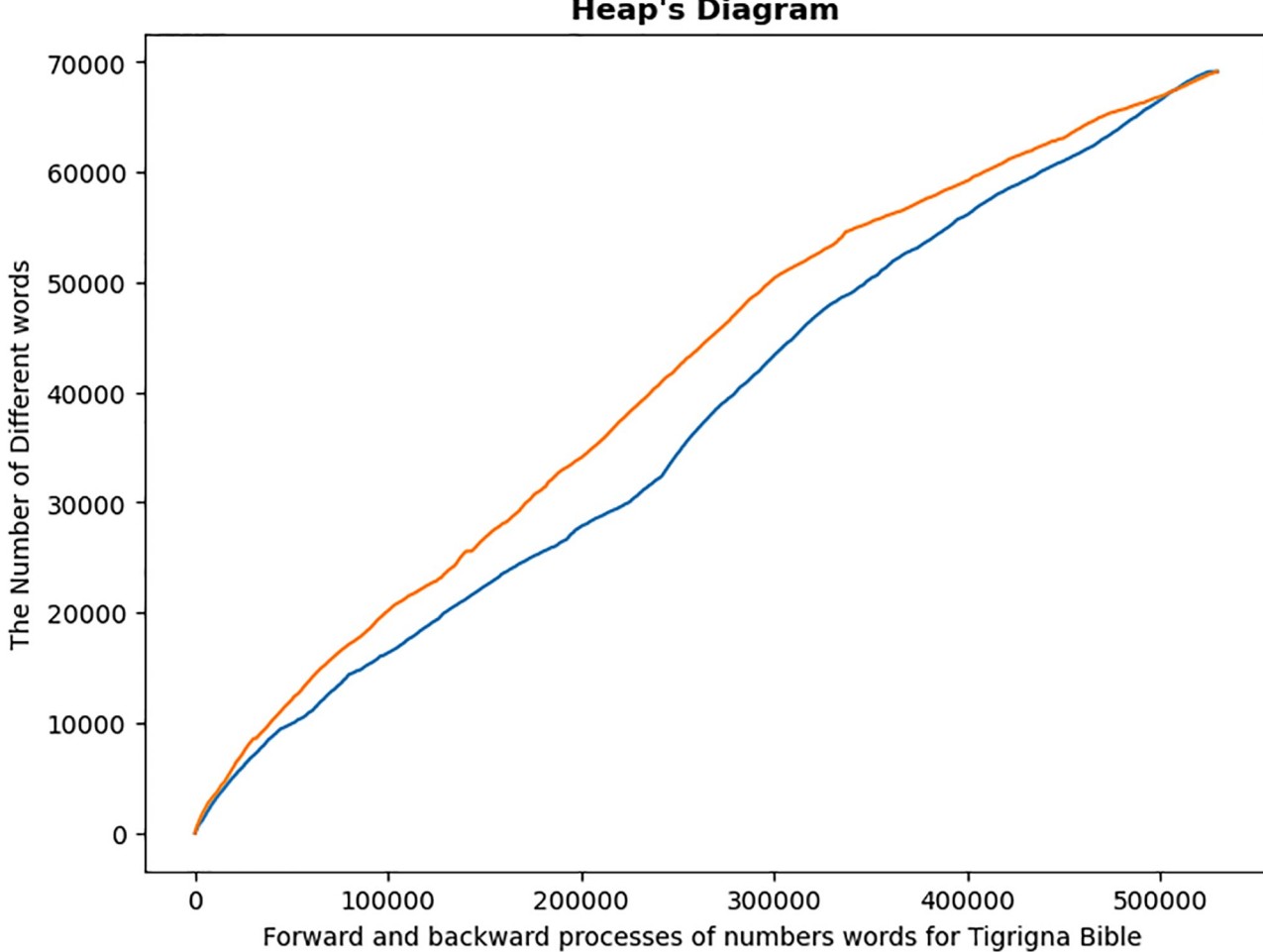

**Fig 3. Forward and backward processes.** The graph of the word count processes vs the number of distinct words in a given text is shown by the blue line, which moves forward, and the red line, which moves backward.

The graph for the process of counting number of different words in forward (blue line) and backward (red line) directions, of Fig 3, did not coincide. Accordingly, it can be inferred that the translated versions of Bible written in the three languages under investigations are not homogeneous within, similar to the analysis of sonnets by [11].

## Parameter estimates

The next two tables provide the findings of parameter estimates for the processes distribution and text homogeneity tests.

The findings revealed that Bible translations of the three languages follows a Zipf's or Zipfian distribution as the value of the parameter $\theta_n$ lies close to 1. The fact that the p-values are almost zero implies the rejection of the null hypothesis. Therefore, it can be concluded that the Bible translations in each of the three languages are not homogeneous. In similar vein as $\omega_n^2$ values are relatively large as compared to the values suggested by [11] for sequences of sonnets from the same author have better homogeneity. Table 1 Using Zipf-Mandelbrot's approach, the estimated values of parameters ($\theta_n$) and ($q_n$) of Eq 3 along with the size of the lexicon or ($n$) and

**Table 1. Zipf-Mandelbrot's parameters.**

| Language | $n$ | $R_n$ | $\theta_n$ | $q_n$ |
|----------|-----|-------|-----------|-------|
| *Tigrigna* | 529114 | 69137 | 0.8829 | -0.8761 |
| *Amharic* | 456823 | 75465 | 0.8267 | 0.9992 |
| *English* | 834243 | 16151 | 0.5514 | 35.523 |

number of different words or vocabulary size ($R_n$) for the Bible in three languages are summarized above. Table 2 The value of the random variable for the scaled difference between counting the number of vocabulary in the forward and backward direction ($J_n$), the estimated values of Zipf-Mandelbrot's parameters in forward ($\theta_n$) and backward ($\widehat{\theta}'_n$), parameter estimate of the variance of the random variable ($J_n$) as well as it's ($p - value$) and Omega square test ($\omega_n^2$).

## 4 Application to Bible classification

Although some Christians think there are 73 books, while others believe there are 81. But the most commonly used is the Bible which is composed of 66 books. Hence, this study is based on the Bible consisting of 66 books that most Christians agree on. The Old Testament, which has 39 books, and the New Testament, which has 27 books, are the two main divisions of the Bible. Since some of the books included are very short in length, Zipf-Mandelbrot law does not fit well. As a result, the researcher decided to use the following classification of Bible into nine genres, which [17] suggested.

1. The Old Testament consists of 39 books which can be subdivide into 5 sub categories:

- Books of the Law (Law)—(Genesis—Deuteronomy 5 books)

- Historical Books (Hist_old)—(Joshua—Esther 12 books)

- Poetic Books—(Job—Songs of Solomon 5 books)

- Major Prophets—(Isaiah—Daniel 5 books)

- Minor Prophets—(Hosea—Malachi 12 books)

2. The New Testament consists of 27 books which can be subdivide into 5 sub categories:

- Gospel—(Matthew—John 4 books)

- Historical Books New (Hist_new)—(Acts 1 book)

- Letters—(Romans—Jude 21 books)

- Book of Vision (Vision)—(Revelations 1 book)

### Text change point detection based on nine genres

Text change point detection,using the commonly accepted classification of the Bible into nine genres according to [17], was performed. The heat maps below show the estimated text point

**Table 2. Parameter estimations and homogeneity tests.**

| Data | $J_n$ | $\theta_n$ | $\theta'_n$ | $\widehat{\theta}_n$ | p-value | $\omega_n^2$ |
|------|-------|-----------|-------------|---------------------|---------|--------------|
| Tigrigna | -15.1545 | 0.8829 | 0.6312 | 0.7571 | 0.0 | 31.417 |
| Amharic | -11.8989 | 0.8267 | 0.6437 | 0.7352 | 0.0 | 8.8030 |
| English | -5.026 | 0.5514 | 0.5330 | 0.5422 | 0.0 | 9.5244 |

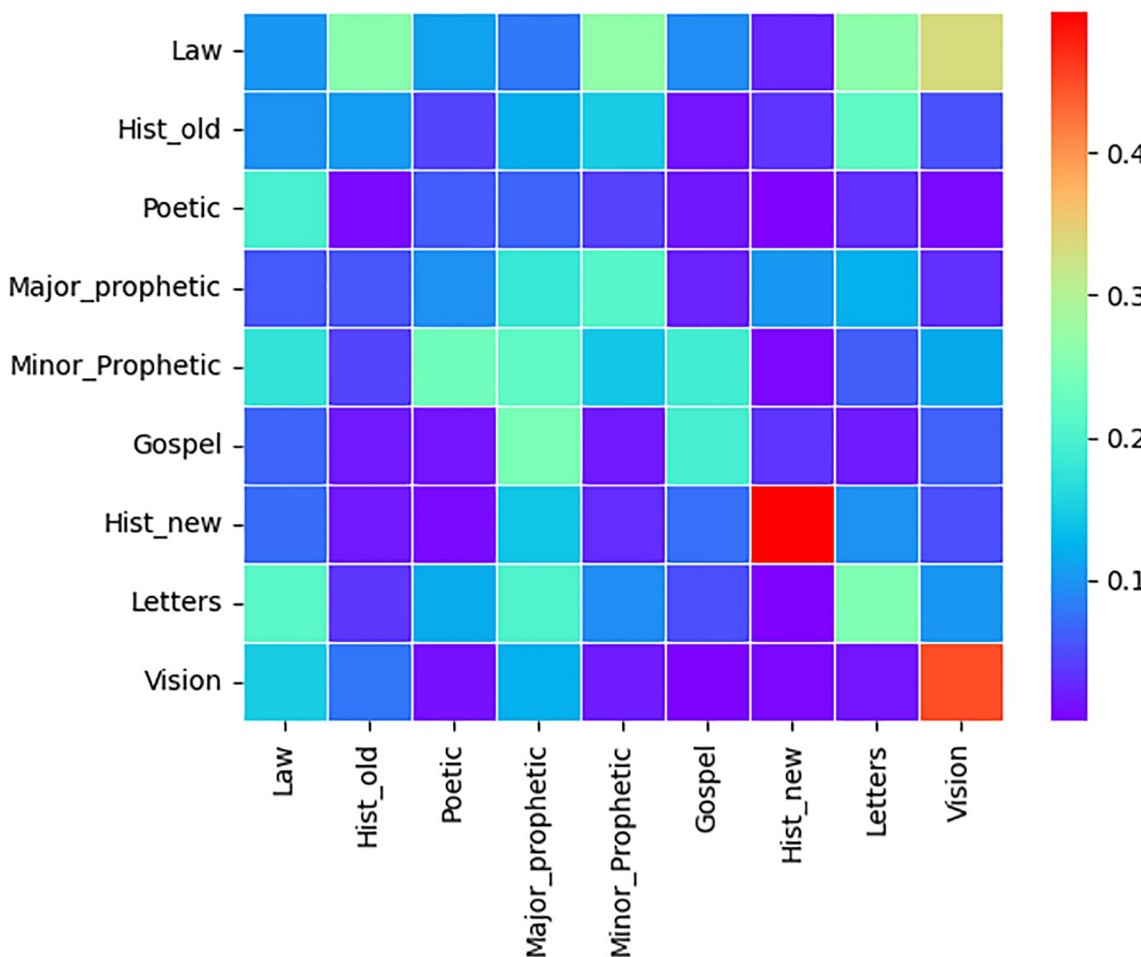

**Fig 4. Tigrigna heat map.** Estimated Error proportion of text change point detection for pairwise genres concatenation of Tigrigna Bible.

change error percentage using Eq 19 for merging two Bible genres (categories) in each language. The error of text point change detection can reached more than 40%. In all the three Heat maps Figs 4–6, the highest error percentages are attained for two Genres Acts (Hist_new) and Revelations (Vision) which are the most homogeneous for each of the three Bible translations. This is due to the fact that each of the two genres have only 1 book. The model can readily recognize the text transition point when the text is a blend of two different papers rather than comparable texts.

According to the findings, when a genre incorporates more than one book, it is a heterogeneous concatenation of various books. A careful examination of one diverse genre, the letters genre in English, yielded the text segmentation results shown below.

## Homogeneity test and text change point detection of Bible letters

The statistical examination of Bible letters in the three languages revealed that they are heterogeneous concatenations of two or more letters. Basically, the Bible letters were divided into two big categories, as follows, and studied individually:

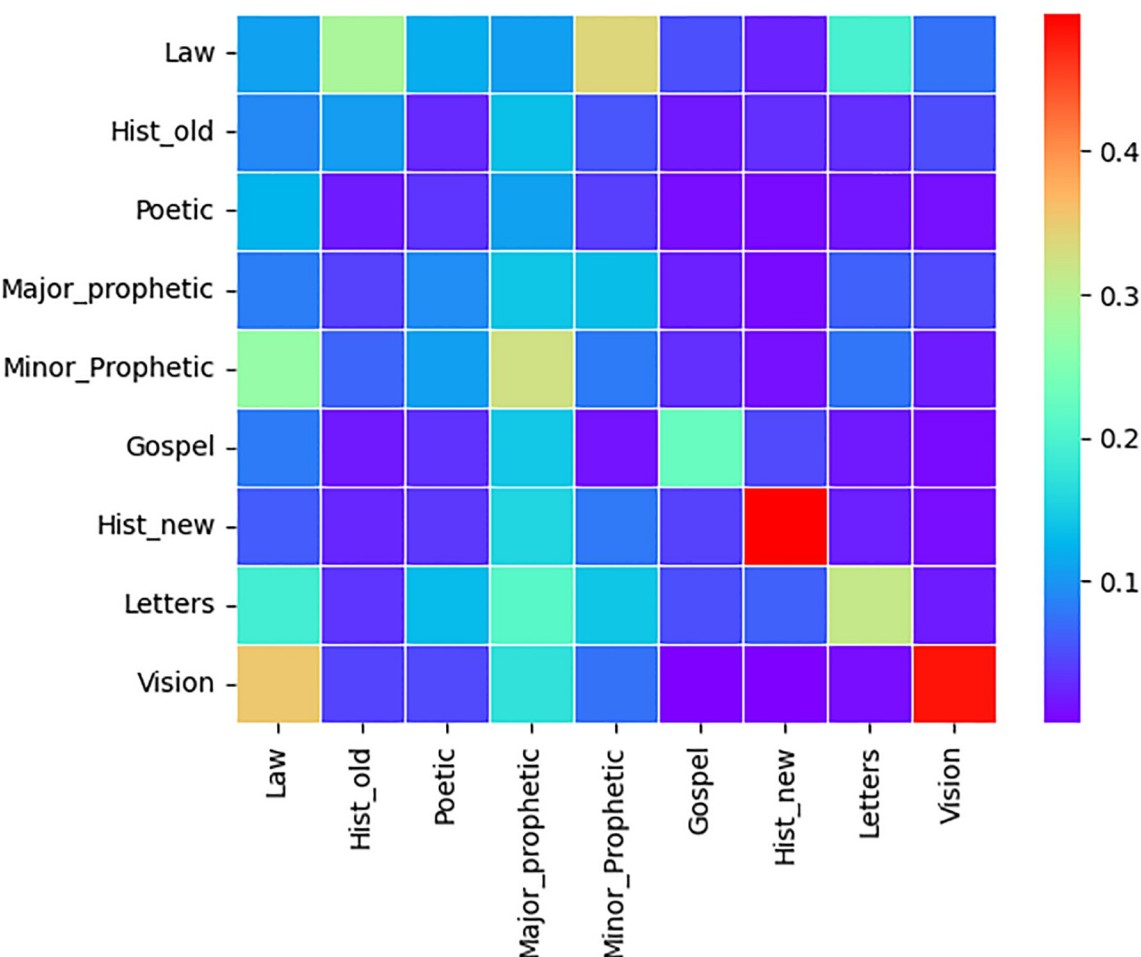

**Fig 5. Amharic heat map.** Estimated Error proportion of text change point detection for pairwise genres concatenation of Amharic Bible.

- Pauline Letters

- Other Letters

Graphical study of Pauline and Other letters revealed that each category is a non-homogeneous mixture of various letters. This can easily be observed from the Heaps diagram for both forward and backward process of distinct word Fig 7. This is to imply that the forward and backward processes did not coincide for each translation.

Graphical analysis, while providing an overall summary of the homogeneity test, is more subjective. As a result, it must be backed up by numerical data. The numerical results are summarized in the following Table 3. The P-values for all six categories are quite small, enabling us to reject the null hypothesis test that the Pauline letters and Other letters are homogeneous texts.

The Other letters in each translation of the Bible have been assumed to have been authored by different authors, indicating their heterogeneity. In-depth research, however is required to comprehend the variability of Pauline letters, notably in the area of text change point identification. As a result, the author undertook extra research on the English Bible's Pauline letters. The first observed text alteration divided the Pauline letters into two groups: Romans to

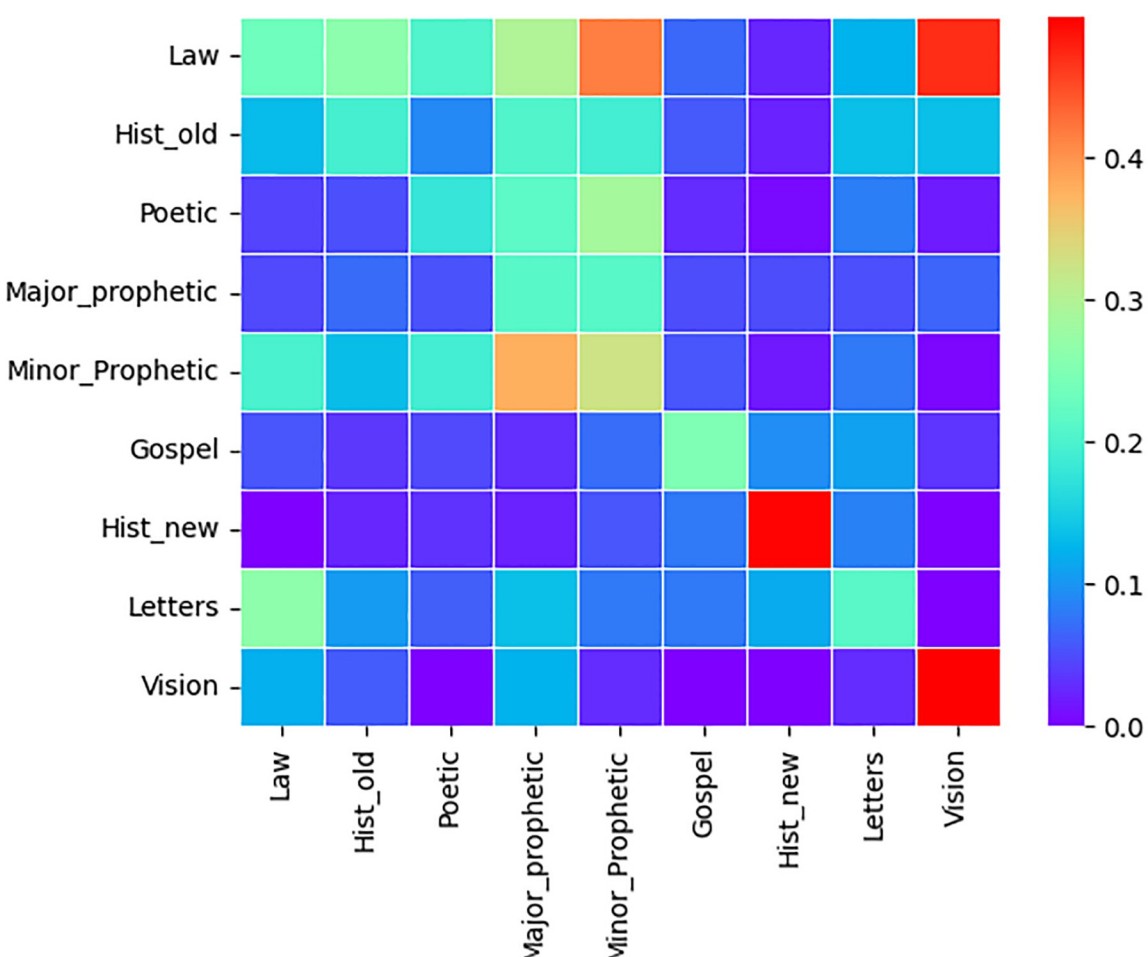

**Fig 6. English heat map.** Estimated Error proportion of text change point detection for pairwise genres concatenation of English Bible.

Philippians and Colossians to Hebrews. A second homogeneity test and text change point detection were performed on the two categories. Text homogeneity tests found that the second group is heterogeneous, whereas the first group has minor homogeneity at 99% but fails at 95% level of confidence.

While the second category is completely heterogeneous (Table 4), a second text change point detection made two segmentation's of Colossians to Hebrews, such as Timothy to Hebrews and Colossians to Thessalonian. Colossians to Thessalonian was found to be statistically significant at 95% but failed at 99%. Whereas Thessalonian to Timothy were not statistically significant for text homogeneous test. Finally, when the Roman to Philippians categories were combined with the Colossians to Thessalonian, the homogeneity result improved. As a result, the English translations of the Pauline letters can be divided into two homogeneous groups:

- Group 1. Romans to Thessalonians letters

- Group 2. Timothy to Hebrews letters

Summarized results of the above analysis are shown in Fig 8 and Table 4 below:

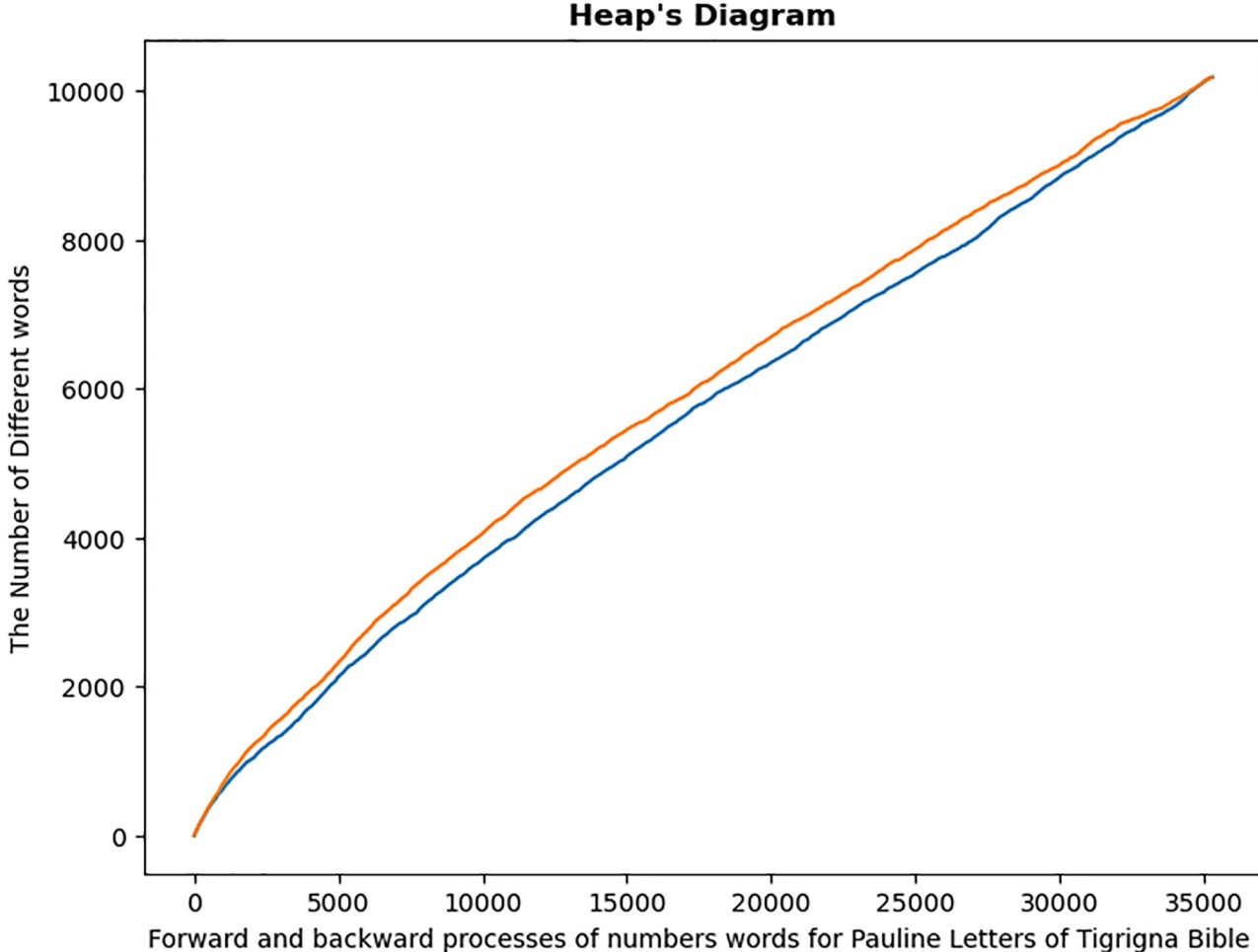

**Fig 7. Heap's diagram of of Paulos and Others letters.** Heaps diagram for the two categories of Bible letters in each translation. Both in forward and backward processes.

**Table 3. Statistical analysis of Pauline letters verses Others in Tigrigna Amharic and English.**

| Parameters | Tigrigna | | Amharic | | English | |
|---|---|---|---|---|---|---|
| $n$ | Paulos | Others | Paulos | Others | Paulos | Others |
| $n$ | 35284 | 7211 | 32032 | 6088 | 54180 | 10659 |
| $R_n$ | 10177 | 2913 | 10218 | 2862 | 3984 | 1627 |
| $\theta_n$ | 0.7770 | 0.8186 | 0.7673 | 0.8240 | 0.5489 | 0.6507 |
| $J_n$ | 0.-2.6290 | 3.1718 | -2.9740 | -2.5831 | -4.3059 | 3.4484 |
| $P-value$ | 0.6.69e-07 | 3.96e-09 | 1.62e-08 | 8.94e-07 | 0.0 | 3.40e-12 |

Table 3 gives number of words, vocabulary size, the value of the random variable for the scaled difference between counting the number of vocabulary in the forward and backward direction ($J_n$), and it's ($p-value$).

**Table 4. Statistical analysis of subdivisions of Paulos letters.**

| Pars | Paulos Letters | | | | |
|---|---|---|---|---|---|
| $n$ | Roman_Phil | Col_Hebr | Col_Thess | Timo_Hebr | Roman_Thess |
| $n$ | 35757 | 18406 | 5249 | 13156 | 41006 |
| $R_n$ | 2986 | 2419 | 973 | 2044 | 3207 |
| $\theta_n$ | 0.5444 | 0.6084 | 0.6249 | 0.6389 | 0.5459 |
| $J_n$ | -1.1834 | -1.9170 | 1.0489 | -0.1549 | -0.5850 |
| $P_{value}$ | 0.0105 | 7.21e-5 | 0.03 | 0.7529 | 0.2064 |

Table 4 gives the number of words, vocabulary size, and the value of the random variable for the scaled difference between counting the number of distinct words in the forward and backward direction ($(J_n)$), and it's ($p - value$).

## 5 Conclusion

In order to widen the scope for these newly established models, the researcher examined recently developed probability models for analysis of text homogeneity and text change point

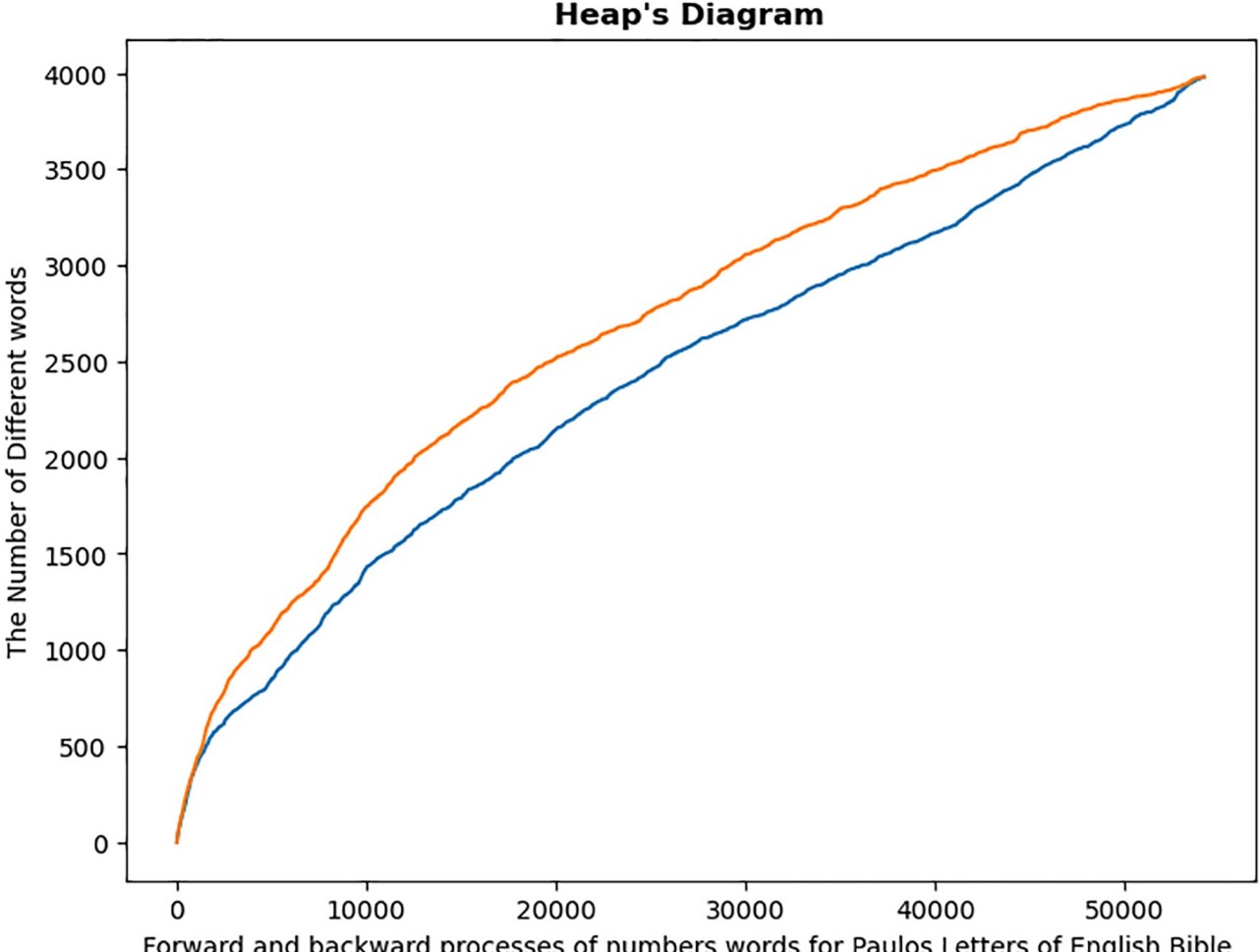

**Fig 8. Heap's diagram of Pauline letters and its subdivisions.** Heaps diagram for forward and backward procedures for Pauline letters and its subdivisions.

detection for Bible studies. Both approaches employ the technique of counting new words forward and backward direction.

The research examined at Bible translations into Tigrigna, Amharic, and English. Each of them containing 66 bookks. Unlike the results obtained for the Quran (Arabic version) by [18], the Heap's diagram, as well as the Zipfian diagram, show that they follow the power function distribution like any other human language [4].

All the Bible translations under investigation results are statistical valid, demonstrating that Bible is a heterogeneous concatenation of many books. Since some of the 66 books are short, the nine genre classification was adapted for text homogeneity test and text segmentation. if a person wishes to deal with short texts, they may need to employ some p-value correction that may be determined by simulating texts using the elementary probability model.

Statistical test of text homogeneity showed that each Bible translation is a heterogeneous concatenations of 9 genres. Seven out of which are heterogeneous and the remaining two are relatively homogeneous.

In addition to testing the homogeneity of a text document, the new technique can be used to trace text change points. The analysis of English version letters genre resulted in two homogeneous segments of the Pauline letters. These new statistical methods can be used to extract words from any language or pseudo-language, which was previously impossible without special training. The disadvantage of this method is that it occasionally generates incorrect results. Hence miscalculating the point of change by a significant proportion of the length is obtained, especially when connecting texts about the same topic or written in a similar style.

## Acknowledgments

The Author would like to thank Professor Artyem Kovaloveskii, my scientific advisor, for his insightful remarks and opinions. I would like also to thank Priest Erimias Biniam for proof reading and editing the language of the manuscript.

## Author Contributions

**Conceptualization:** Berhane Abebe.

**Data curation:** Berhane Abebe.

**Methodology:** Berhane Abebe.

**Project administration:** Berhane Abebe.

**Software:** Berhane Abebe.

**Writing – original draft:** Berhane Abebe.

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
