## [Decision Letter · Decision Letter 0]

15 Nov 2023

PONE-D-23-20994HOMOGENEITY AND CHANGE POINT DETECTION USING FORWARD AND BACKWARD PROCESSES: FOR BIBLE TEXTSPLOS ONE

Dear Dr. Andemichael,

Thank you for submitting your manuscript to PLOS ONE. After careful consideration, we feel that it has merit but does not fully meet PLOS ONE’s publication criteria as it currently stands. Therefore, we invite you to submit a revised version of the manuscript that addresses the points raised during the review process.

We look forward to receiving your revised manuscript.

Kind regards,

Roy Cerqueti, Ph.D.

Academic Editor

PLOS ONE

Journal Requirements:

3. Please upload a new copy of Figures 1 to 5 as the detail is not clear. Please follow the link for more information: 

https://blogs.plos.org/plos/2019/06/looking-good-tips-for-creating-your-plos-figures-graphics/
https://blogs.plos.org/plos/2019/06/looking-good-tips-for-creating-your-plos-figures-graphics/

4. Please ensure that you refer to Figures 6 to 8 in your text as, if accepted, production will need this reference to link the reader to the figure.

5. We note you have included a table to which you do not refer in the text of your manuscript. Please ensure that you refer to Table 3 in your text; if accepted, production will need this reference to link the reader to the Table.

**Additional Editor Comments:**

Dear author, The paper has an interesting research question, but it it not publishable in its current form. The reviewers have some doubts about the effectiveness of a revision round, but I want to give you the opportunity to revise deeply the paper.  You should convince me that the methodological section and the discussion of the results have a satisfactorily high level to grant the publication in Plos One. I would say that this is a risky major revision. Please, read carefully the comments of the revievers. Thank you for submitting your paper to Plos One.

Reviewers' comments:

Reviewer's Responses to Questions

**Comments to the Author**

1. Is the manuscript technically sound, and do the data support the conclusions?

Reviewer #1: Partly

Reviewer #2: Yes

2. Has the statistical analysis been performed appropriately and rigorously? 

Reviewer #1: Yes

Reviewer #2: No

3. Have the authors made all data underlying the findings in their manuscript fully available?

Reviewer #1: Yes

Reviewer #2: Yes

4. Is the manuscript presented in an intelligible fashion and written in standard English?

Reviewer #1: No

Reviewer #2: No

5. Review Comments to the Author

Reviewer #1: Minor points:

1- The manuscript needed to be reread since there is a considerable number of typos and incorrect grammar throughout the text. I list some of the language problems in the following:

• Abstract

Zifp-Mandelbrot  Zipf-Mandelbrot

• Introduction

therefore,expected  therefore, expected

Bahdur  Bahadur

mendelbrot  Mandelbrot

homogeniety  homogeneity

• Theoretical Background

Hurvitz zeta function  Hurwitz zeta function

number of word  number of words

• Bible Classification

old testament  Old Testament

( Job  (Job

new Testament  New Testament

• Application to Bible Books in 3 Languages

more then 600 words  more than 600 words

2- Citations in the text are started from [16]. They should be sorted in ascending order: [1], [2], [3], … .

3- Text should be justified.

4- What is [?] in lines 11 and 12 of page 2/8.

5- All formulas should be labeled by numbers.

6- All parameters and variables in formulas have to be introduced in the text.

7- P.5/8, L.84 & L.85: Short paragraphs and unintelligible sentences.

8- It will be illustrative if author give an example for calculation of parameters in table 3.

9- Fig.4 and Fig.5 should be explained clearly in the text. Moreover, red and blue colors have to be indicated in fig.4 and fig.5.

10- All references have to be written in the same style. Please take a look at references 8, 9, 11, …. In some references authors name is written completely and in some others just the first letter of their name written.

Major concern:

1- P.4/8, L.47: Author has to explain Table (and the next tables) completely. Its all parameters and values should be clarified in detail.

2- Heat maps are not properly described. Author has to clearly explain what they refer to.

3- All parameters in tables have to be determined clearly in the captions.

4- P.6/8, L.96: Table does not have caption! What is the third column in this table?

5- Please explain why “longer texts consistently have lower error rates”.

6- Forward and backward processes should be described in detail.

Reviewer #2: The paper "Homogeneity and change point detection using forward and backward processes: for Bible texts" analyses the text of the Bible written in three different languages (Tigrigna, Amharic, and English) and shows how they meet the requirements of texts written in human language, how they are very heterogeneous within them and tests possible change points from one text model to another.

The premises and the initial idea of the paper are interesting, but the part of the empirical analysis is meagre and needs to be completely argued. The methodology section is also a simple list of formulas left somewhat to itself and is quite unrelated from the empirical analysis. Unfortunately on reading, the paper appears sloppy in all its parts.

Wishing to reward the initial idea and analytical effort that is not accompanied by adequate interpretation or commentary, I suggest that the paper be heavily revised in all its parts. Only in this case the paper can be considered for publication.

I also suggest a final revision of the English.

Some minor but equally important points:

1) The title should be made more intriguing

2) The Theoretical Background section should actually be called Methodology

3) Beware of copy and paste. One sentence is repeated twice (pp 6 and 7)

4) Beware of figures and their order. Figures 7 and 8 are both labelled as Figure 7, also the figures should be ordered from first to last.

6. PLOS authors have the option to publish the peer review history of their article (what does this mean?). If published, this will include your full peer review and any attached files.

Reviewer #1: No

Reviewer #2: No

---

## [Author Response · Author response to Decision Letter 0]

21 Dec 2023

Thanks Very Much for the editor as well as the reviewers for their constructive comments and suggestions. The manuscript was revised thoroughly taking into account each and every comment and suggestion. But due to extensive revision some of them might not have been included in the track change manuscript.

---

## [Decision Letter · Decision Letter 1]

11 Apr 2024

PONE-D-23-20994R1Application of elementary probability models for text homogeneity and segmentation: a case study of BiblePLOS ONE

Dear Dr. Andemichael,

Thank you for submitting your manuscript to PLOS ONE. After careful consideration, we feel that it has merit but does not fully meet PLOS ONE’s publication criteria as it currently stands. Therefore, we invite you to submit a revised version of the manuscript that addresses the points raised during the review process. Please see comments from the journal office below.

We look forward to receiving your revised manuscript.

Kind regards,

Roy Cerqueti, Ph.D.

Academic Editor

PLOS ONE

Journal Requirements:

Additional Editor Comments:

Note from Journal Staff: In order to meet PLOS ONE publication criteria please ensure the manuscript has been copyedited, and please address the following:

1) Please include supporting citations for Lines 171 - 176;

2) typo on line 244;

3) grammatical error in line 247. 

Reviewers' comments:

Reviewer's Responses to Questions

**Comments to the Author**

1. If the authors have adequately addressed your comments raised in a previous round of review and you feel that this manuscript is now acceptable for publication, you may indicate that here to bypass the “Comments to the Author” section, enter your conflict of interest statement in the “Confidential to Editor” section, and submit your "Accept" recommendation.

Reviewer #1: All comments have been addressed

2. Is the manuscript technically sound, and do the data support the conclusions?

Reviewer #1: Yes

3. Has the statistical analysis been performed appropriately and rigorously? 

Reviewer #1: Yes

4. Have the authors made all data underlying the findings in their manuscript fully available?

Reviewer #1: Yes

5. Is the manuscript presented in an intelligible fashion and written in standard English?

Reviewer #1: Yes

6. Review Comments to the Author

Reviewer #1: (No Response)

7. PLOS authors have the option to publish the peer review history of their article (what does this mean?). If published, this will include your full peer review and any attached files.

Reviewer #1: No

---

## [Author Response · Author response to Decision Letter 1]

14 Apr 2024

Thanks Very Much. I really appreciate all the comments of the editor have been considered.

---

## [Editor Report · Decision Letter 2]

25 Apr 2024

Application of elementary probability models for text homogeneity and segmentation: a case study of Bible

PONE-D-23-20994R2

Dear Dr. Andemichael,

We’re pleased to inform you that your manuscript has been judged scientifically suitable for publication and will be formally accepted for publication once it meets all outstanding technical requirements.

Kind regards,

Roy Cerqueti, Ph.D.

Academic Editor

PLOS ONE
---

## [Editor Report · Acceptance letter]

3 May 2024

PONE-D-23-20994R2 

PLOS ONE

Dear Dr. Abebe, 

I'm pleased to inform you that your manuscript has been deemed suitable for publication in PLOS ONE. Congratulations! Your manuscript is now being handed over to our production team.

Kind regards, 

on behalf of

Professor Roy Cerqueti 

Academic Editor

PLOS ONE